# N58A Exerts Analgesic Effect on Trigeminal Neuralgia by Regulating the MAPK Pathway and Tetrodotoxin-Resistant Sodium Channel

**DOI:** 10.3390/toxins13050357

**Published:** 2021-05-17

**Authors:** Chun-Li Li, Ran Yang, Yang Sun, Yuan Feng, Yong-Bo Song

**Affiliations:** 1Department of Pharmacology, Shenyang Pharmaceutical University, Shenyang 110016, China; lichunli@syphu.edu.cn (C.-L.L.); yangran_ran123@126.com (R.Y.); sunyang_2021@126.com (Y.S.); alny_feng@sina.com (Y.F.); 2School of Life Sciences and Biopharmaceutical Science, Shenyang Pharmaceutical University, Shenyang 110016, China

**Keywords:** trigeminal neuralgia, N58A, Nav1.8, Nav1.9, MAPK, analgesic effect

## Abstract

The primary studies have shown that scorpion analgesic peptide N58A has a significant effect on voltage-gated sodium channels (VGSCs) and plays an important role in neuropathic pain. The purpose of this study was to investigate the analgesic effect of N58A on trigeminal neuralgia (TN) and its possible mechanism. The results showed that N58A could significantly increase the threshold of mechanical pain and thermal pain and inhibit the spontaneous asymmetric scratching behavior of rats. Western blotting results showed that N58A could significantly reduce the protein phosphorylation level of ERK1/2, P38, JNK, and ERK5/CREB pathways and the expression of Nav1.8 and Nav1.9 proteins in a dose-dependent manner. The changes in current and kinetic characteristics of Nav1.8 and Nav1.9 channels in TG neurons were detected by the whole-cell patch clamp technique. The results showed that N58A significantly decreased the current density of Nav1.8 and Nav1.9 in model rats, and shifted the activation curve to hyperpolarization and the inactivation curve to depolarization. In conclusion, the analgesic effect of N58A on the chronic constriction injury of the infraorbital (IoN-CCI) model rats may be closely related to the regulation of the MAPK pathway and Nav1.8 and Nav1.9 sodium channels.

## 1. Introduction

Trigeminal neuralgia (TN) is a clinically typical chronic neurological disease. The International Association for the Study of Pain (IASP) defines it as sudden, severe, transient, severe, often unilateral, acupuncture-like, recurrent neuropathic pain in one or more branches of the trigeminal nerve. It can be caused by a light touch on the face, such as touching, eating, talking, washing the face, or slight irritation such as natural wind [1]. The onset of trigeminal neuralgia may last for several days, months, or even years, which has a significant impact on the patient’s daily life and work quality. However, at present, there is no ideal treatment method in clinical practice.

VGSC is closely related to neuropathic pain and inflammatory pain. According to the difference of the α subunit structure, VGSC can be divided into nine subtypes (Nav1.1–Nav1.9). The subtypes of VGSC can be divided into two types according to their sensitivity to tetrodotoxin (TTX). Nav1.1–1.4, Nav1.6, and Nav1.7 are sensitive to TTX; that is, the maximum conduction block to TTX occurs between 1 and 25 nM [2]. Nav1.5, Nav1.8, and Nav1.9 are resistant to TTX and need more than 1 µM of TTX to block half of the largest channels. All human VGSC subtypes have more than 75% amino acid sequence identity, and each VGSC subtype is specifically distributed in cells and tissues, indicating that it can be studied by anesthesia or selecting a specific region of VGSC. Sodium channels are closely related to neuropathic pain and inflammatory pain. It has been confirmed that among the VGSC subtypes, Nav1.3, Nav1.7, Nav1.8, and Nav1.9 play an important role in the generation and transmission of pain signals. VGSC is involved in the transmission of pain stimuli in the sensory nerves, suggesting its great potential in the development of new analgesics.

Mitogen-activated protein kinase (MAPK) is a conserved tertiary kinase response pathway involved in intracellular signal transduction, cell proliferation, and differentiation. The MAPK family contains four different signal pathways: ERK1/2, P38MAPK, JNK, and ERK5 [3]. Studies have shown that the activation of the MAPK pathway in the primary afferent nerve and spinal cord contributes to pain hypersensitivity and changes in neuronal plasticity, and is closely related to neuropathic pain and inflammatory pain [4]. The ERK5/CREB pathway plays a key role in pain signal transduction and leads to pain hypersensitivity. In addition, CREB-dependent gene expression is believed to be related to central sensitization associated with persistent pain states [5]. These findings indicate that the ERK5/CREB pathway plays a vital role in both inflammatory pain and neuropathic pain.

As a classical opioid analgesic, morphine is often used for severe pain such as cancer pain. However, it has side effects such as drug tolerance, addiction, respiratory inhibition, gastrointestinal reaction, and so on, which greatly limits the clinical application of morphine [6]. Scorpion analgesic peptide N58A is a β-type scorpion neurotoxin isolated and purified from East Asian scorpion venom. According to the cDNA sequence of N58A, the research group of Professor Zhang of Shenyang Pharmaceutical University used DNA recombinant technology and PCR amplification technology to purify and express in *E**. coli* with a purity of more than 90%. The N58A protein mutant designed by protein engineering technology has a molecular weight of about 9 KDa and a UV absorption peak of 279.4 nm (Appendix A. Previous studies have confirmed that the unique structure of N58 plays a vital role in the activity of toxins and significantly inhibits sodium ion currents [7]. Experiments have confirmed that N58A has significant analgesic and anti-inflammatory effects on rats, and no neurotoxicity is observed within the effective analgesic dose range.

In summary, this study took the IoN-CCI model as the research object. Through behavioral, molecular biology, and electrophysiological experiments, the analgesic effect of N58A on trigeminal neuralgia, the effect of IoN-CCI surgery on Nav1.8 and Nav1.9 in rats’ TG cells, and the protein phosphorylation level of the MAPK pathway all of these things were investigated, in order to provide a theoretical basis for further elucidating the analgesic effect of N58A and its possible mechanism.

## 2. Results

### 2.1. The Spatial Conformation of N58A

The double-precision MD simulation method of Amber18 was used to clarify the spatial conformation of N58A in a real physiological environment (Figure 1, Appendix A).

### 2.2. Behavioral Results

On the seventh day after the establishment of the model, the thermal pain threshold and mechanical pain threshold of the right face of rats decreased significantly, and decreased to the lowest on the 14th day. Therefore, N58A was injected into the caudal vein on the 14th day after operation. Compared with the model group, the middle-dose group and high-dose group showed significant analgesic effect 0.5 h after a single dose of N58A, and in a dose-dependent manner. The maximum analgesic effect appeared at about 2 h and lasted for 4 h, and its analgesic effect was similar to that of morphine. No significant changes were observed in the non-operative side before and after administration, but the mechanical pain threshold increased to a certain extent at 4 h of administration (Figure 2a–d). Meanwhile, N58A significantly inhibited spontaneous asymmetric face scratching in rats in a dose-dependent and time-dependent manner. That is, the scorpion analgesic peptides in the middle-dose and high-dose groups showed better analgesic effect, and their analgesic effect was similar to that of morphine (Figure 2e).

### 2.3. The Effects of N58A on Protein Phosphorylation of MAPK and ERK5/CREB Pathway and Expression of Nav1.8 and Nav1.9 Proteins

The phosphorylation levels of ERK1/2 protein, ERK5 protein, CREB protein, JNK protein, and P38 protein in the trigeminal ganglion neurons in the model group were significantly higher than those in the sham group. However, the phosphorylation levels of ERK1/2 protein, ERK5 protein, CREB protein, JNK protein, and P38 protein in each dose group of the N58A group were lower than that of the model group. The results showed that N58A could significantly inhibit the phosphorylation of ERK1/2, ERK5, CREB, JNK, and P38 proteins in a dose-dependent manner (Figure 3a–e). After the IoN-CCI model, the expression of Nav1.8 and Nav1.9 protein in trigeminal ganglion neurons decreased significantly. Caudal intravenous injection of morphine had no significant effect on protein expression. After administration of N58A, the relative expression of Nav1.8 protein in the 2.0 mg/kg group and the 4.0 mg/kg group was significantly lower than that in the model group, and the relative expression of Nav1.9 protein in the 4.0 mg/kg group was significantly lower than that in the model group. The results showed that N58A could inhibit the expression of Nav1.8 and Nav1.9 proteins in a dose-dependent manner (Figure 3f,g).

### 2.4. Gait Analysis

Gait analysis training was performed seven days before surgery. On the 13th day after the establishment of the model, the width ratio of the forefoot touchdown support in the model group was lower than that in the sham group to a certain extent, but on the 14th day after administration, the width of the forefoot support in the model group returned to the same level as that in the sham group. In addition, there was no significant difference in the width of ground support between the sham group and the sham administration group at each time point (Figure 4a). At the same time, 13 days after the establishment of the IoN-CCI model, the movement frequency of the forelimb was significantly lower than that of the hindlimb. However, after administration of N58A, the forelimb movement frequency in rats returned to the same level as that in the sham group and before the establishment of the model. However, there was no change in the sham administration group compared with the sham group, indicating that N58A had no effect on the motion symmetry coefficient of the sham group (Figure 4b). Similarly, on the 13th day after the establishment of the model, it was obvious that the limb swing time of the surgical forelimb of the rats in the administration group increased, indicating that the forelimb motility of the rats on the surgical side became worse, but this change disappeared immediately after administration, which proved that this effect was completely caused by pain and there was no organic injury of the limbs. On the 14th day, there was no significant difference between the sham administration group and the sham group, which proved that caudal vein administration of N58A did not interfere with the control of limb movement in rats (Figure 4c).

### 2.5. Electrophysiological Results

#### 2.5.1. The Effects of N58A on the Amplitudes of Nav1.8 and Nav1.9 Currents in TG Neurons of IoN-CCI Rats

To study the effect of N58A on the peak current density of Nav1.8, Nav1.8 currents were recorded when the cells were clamped at –110 mV first, and then depolarized to –50 mV. The depolarization step from –50 mV to +50 mV was given at 500 ms, and the peak current of Nav1.8 was recorded and counted. After IoN-CCI surgery, the peak sodium current density mediated by Nav1.8 was significantly decreased by 51.1%. Compared to the model group, morphine group had no significant change in the Nav1.8 peak current density, and the Nav1.8 peak current density decreased significantly by 48.0% after the administration of 4.0 mg/kg N58A (Figure 5a,b).

In order to study the effect of N58A on the Nav1.9 peak current density, the Nav1.9 current was recorded when the cells were clamped at –100 mV, and then the depolarization step from –80 mV to –35 mV was given, from which the Nav1.9 peak current could be recorded and analyzed. The results showed that the Nav1.9 peak current density of the model group was significantly lower than that of the sham group. Compared to the model group, the morphine group had no significant effect on the current density of Nav1.9, but the peak current density of Nav1.9 decreased significantly after the administration of N58A at 4.0 mg/kg (Figure 5c,d).

#### 2.5.2. The Effects of N58A on I-V and Steady-State Activation Curves of Nav1.8 and Nav1.9 Currents

The results showed that compared with the sham group, the I-V curve of the Nav1.8 current in the model group shifted significantly, and the peak current activation voltage shifted from –10 mV to 0 mV. Compared to the model group, the morphine group had no significant effect on the I-V curve of Nav1.8 current. After the administration of N58A, the I-V curve of the Nav1.8 current continued to move up, and the peak current activation voltage decreased back to –10 mV (Figure 6a,b)

In the Nav1.9 current I-V curve, the I-V curve of the IoN-CCI group was significantly higher than that of the sham group. Compared to the model group, the morphine group had no significant effect on the I-V curve, whereas the N58A group showed a slight upward shift in the I-V curve (Figure 6c,d).

According to the above protocol, a series of Nav1.8 and Nav1.9 currents were recorded. The results showed that compared to the sham group, the steady-state activation curve of the Nav1.8 current in the model group shifted significantly in the direction of depolarization. Compared to the model group, there was no significant effect on the morphine group, but the N58A group drifted to hyperpolarization. The *V*_1/2_ and slope factor *k* of the control group, sham group, IoN-CCI group, IoN-CCI + Mor group, and IoN-CCI + N58A group were –16.46 ± 0.38, 2.40 ± 0.22, 16.25 ± 0.88, 2.71 ± 0.46, 10.91 ± 1.16, 3.28 ± 0.45, 9.90 ± 0.67, 3.97 ± 0.48, 19.88 ± 1.06, and 6.53 ± 0.51, respectively (Figure 6e). The steady-state activation curve of Nav1.9 was drawn by fitting the Boltzmann equation. The results showed that compared to the sham group, the steady-state activation curve of the Nav1.9 current in the IoN-CCI group shifted in the direction of depolarization. Compared to the model group, there was no significant effect in the morphine group, but the N58A group drifted to hyperpolarization. The half activation voltage *V*_1/2_ and slope factor *k* of each group were –40.16 ± 0.26, 5.01 ± 0.12, 40.45 ± 0.32, 5.28 ± 0.15, 37.89 ± 0.71, 5.19 ± 0.26, 35.61 ± 1.13, 5.91 ± 0.32, 43.76 ± 0.22, 4.63 ± 0.19, respectively (Figure 6F).

#### 2.5.3. The Effects of N58A on Steady-State Inactivation Curves of Nav1.8 and Nav1.9

The results showed that compared to the sham group, the Nav1.8 current inactivation curve of the IoN-CCI group shifted in the direction of hyperpolarization. Compared to the model group, there was no significant change in the morphine group, and the Nav1.8 current inactivation curve of the N58A group shifted in the direction of depolarization. The values of the parameters of *V*_1/2_ and slope factor *k* of the control group, sham group, IoN-CCI group, IoN-CCI + Mor group, and IoN-CCI + N58A group were –19.51 ± 1.03, 6.17 ± 0.92, 18.61 ± 0.63, 5.39 ± 0.56, 27.38 ± 1.25, 5.98 ± 1.08, 6.92 ± 1.09, 26.16 ± 0.52, 4.62 ± 0.42, respectively (Figure 7a,b).

In the inactivation current of Nav1.9, compared to the sham group, the surgical inactivation curve of the IoN-CCI group drifted in the direction of hyperpolarization. Compared to the model group, there was no significant change in the morphine group, and the Nav1.9 current inactivation curve of the N58A group drifted slightly in the depolarization direction. The values of the parameters of *V*_1/2_ and slope factor *k* of each group were –77.34 ± 0.88, 7.40 ± 0.57, 76.90 ± 0.75, 7.74 ± 0.30, 83.36 ± 0.33, 5.73 ± 0.29, 84.15 ± 0.39, 6.11 ± 0.33, 82.64 ± 0.68, 6.36 ± 0.45, respectively (Figure 7c,d).

## 3. Discussion

Peripheral nerve demyelination and vascular lesions are the basis of TN. According to vascular compression theory and demyelination theory, the IoN-CCI model can produce significant and stable spontaneous pain after operation [8], so it is widely used in the study of trigeminal neuralgia. In addition, the IoN-CCI model belongs to a chronic neuropathic pain model, which lasts for a long time and can be used to study the changes of pain in different time periods [9]. In this experiment, the threshold of thermal pain and mechanical pain decreased significantly five to seven days after the IoN-CCI operation, and could reduce spontaneous facial scratching and other autonomic behaviors, indicating that thermal hyperalgesia and mechanical hyperalgesia were successfully induced in rats, and the model was established successfully.

In order to evaluate the effect of N58A on rat behavior, we used a gait analysis system. Among the many evaluation items, some evaluation items related to pain or nerve injury and motor ataxia were selected. The paw area can indicate the maximum contact area of the paws that touch the ground when the rat is traveling. The maximum paw area is used to evaluate the load capacity and pain degree of the animal’s limbs to assess the degree of related nerve damage in the rat. The changes before and after this data can reflect the postural adjustments made by the animals for stability. Existing studies have shown that cocaine administration during the fetal period narrows the stride of rats during growth [10]. In this numerical value, the width of the forelimb of healthy rats will be less than that of the hind limbs, and the support basis also depends to a large extent on the body size, so in the course of the experiment, rats with the same body length should be selected as the experimental subjects. Gait symmetry represents the ratio of the forelimb step frequency to the hind limb step frequency. For a healthy animal that can only act flexibly, the motion symmetry coefficient is the same, and the motion symmetry coefficient of healthy rats is 1. It has been reported that in animals with spinal cord injury, the frequency of posterior limb steps increases to make up for the insufficiency of the hind limbs, in which case the coefficient of symmetry deviates from 1 [11]. At the same time, the motor symmetry of rats decreases with age. Swing duration describes the duration of the limb wobble when the animal’s claws fall to the ground. This value can be used to evaluate the flexibility of animal limb movement. Arthritic animals extend the swing time due to the loss of limb joint mobility, as well as rats with limb pain [12]. The changes in the tested rats in the above items reflected the effect of N58A on the motor coordination ability of rats, and the results were obtained according to the data changes in the sham operation and model groups before and after administration. The single dose of N58A 4.0 mg/kg did not affect the weight-bearing ability of limbs, the flexibility of joints, or the motor coordination of limbs in rats, so it can be considered that it will not lead to ataxia in rats.

The MAPK pathway is closely related to the regulation of neuropathic pain and inflammation [13]. P38MAPK is mainly involved in mature nervous system inflammation and hyperalgesia. Inhibitors of the P38MAPK pathway can reduce the abnormal excitation of neurons under neuropathic pain [14]. Intrathecal injection of a P38 inhibitor can increase the pain threshold of neuropathic pain in various animal models, which suggests that the activation of the P38 pathway is closely related to neuropathic pain [15]. ERK1/2 can be activated in microglia or astrocytes and plays a key role in the process of spinal dorsal horn pain sensitization [16]. At the same time, it also participates in the transmission and regulation of pain signals, and plays an important role in the occurrence and maintenance of inflammatory pain and neuropathic pain. Nerve injury leads to the synthesis of inflammatory or nociceptive mediators, which can induce DRG neurons to produce rapid activation of the JNK pathway, thus enhancing and prolonging pain signal transduction. Intrathecal injection of the JNK inhibitor can relieve neuropathic pain in an SNL model [17]. Therefore, the activated JNK pathway plays an important role in the maintenance of neuropathic pain. ERK5 is a new member of the MAPK family. ERK5 is activated in spinal nerve injury. Intrathecal injection of ERK5 antisense oligodeoxynucleotides can alleviate neuropathic pain [18]. As the downstream target of ERK5, CREB plays a key role in the occurrence and development of neuropathic pain by regulating the transcription and secretion of a variety of neurotransmitters.

Some studies have shown that MAPKs plays an important role in the excitation of sensory neurons, and the expression of MAPK (ERK1/2, P38) and sodium channels (Nav1.7, Nav1.8) are upregulated in human neuroma [19]. In the rat model of nerve injury, the expression of P38 was upregulated, and the current densities of TTX-R and TTX-S increased in DRG neurons [20]. The activation of P38 can change the current density of Nav1.8 and Nav1.6, but does not change the gating characteristics of the channel [21]. ERK1/2 could induce the activation and inactivation curve of Nav1.7 in DRG neurons to shift to hyperpolarization and increase the frequency of action potential emission, but had no effect on current density [22]. Therefore, we believe that MAPKs may preferentially target specific sodium channel subtypes and differentially regulate channel conductance and gating properties. In this study, we took the IoN-CCI model as the research object to investigate the changes in the protein phosphorylation level of the MAPK pathway after modeling and administration. The results showed that after the establishment of the model, the phosphorylation levels of ERK1/2, JNK, P38, and ERK5/CREB proteins increased significantly, whereas the protein phosphorylation levels decreased significantly after administration in a dose-dependent manner. It is suggested that N58A may exert its analgesic effect by inhibiting the hyperalgesia caused by central sensitization of TN through the ERK1/2, P38, JNK, and ERK5/CREB pathways.

At present, in the investigation of various neuropathic pain models, the protein expression results of Nav1.8 and Nav1.9 are not completely consistent, and the upregulation or downregulation of protein expression have been reported. In this study, it was found that Nav1.8 and Nav1.9 were significantly downregulated in the IoN-CCI model rats, so it is inferred that the pathogenesis of neuropathic pain may be closely related to the downregulation of Nav1.8 and Nav1.9. Further downregulation of Nav1.8 and Nav1.9 was found after the administration of N58A, indicating that N58A may exert an analgesic effect by inhibiting the function and protein expression of Nav1.8 and Nav1.9. However, how it plays a role in pain needs to be further explored.

The pain caused by nerve injury is closely related to Nav1.8 and Nav1.9 channels. In a variety of neuropathic pain and inflammatory pain models, Nav1.8 is an important part of the ascending branch of action potential formation [23], whereas Nav1.9 plays an important role in the formation of resting potential [24]. In this experiment, we investigated the function of Nav1.8 and Nav1.9 after IoN-CCI modeling and administration. The results showed that after the CCI model, the current densities of Nav1.8 and Nav1.9 decreased in varying degrees. It has been reported that nerve injury can downregulate the expression of Nav1.8 mRNA and protein in DRG neurons [25], suggesting that the decrease in current density of Nav1.8 and Nav1.9 may be due to the decreased expression of Nav1.8 and Nav1.9 protein in TG neurons, which is consistent with the results of the WB experiment. When N58A was given, the current density also decreased to different degrees, which preliminarily proved that N58A has different inhibitory effects on Nav1.8 and Nav1.9.

Scorpion analgesic peptide N58A is a β-type toxin that can play a role by regulating the activation and inactivation of voltage-gated sodium channels. In this experiment, the kinetic characteristics of Nav1.8 and Nav1.9 after CCI modeling and administration were investigated. The results showed that compared to the sham group, the steady-state activation curve of Nav1.8 after the CCI model shifted to the right and the steady-state inactivation curve shifted to the left. It is suggested that the hyperalgesia caused by the model may accelerate the inactivation of the channel and accelerate it into the state of resurrection, thus increasing the discharge frequency of neurons and improving the excitability of neurons. After administration of N58A, it was found that the steady-state activation curve of Nav1.8 drifted to the left and the steady-state inactivation curve drifted to the right. It is suggested that the analgesic effect of N58A may reverse the abnormal discharge frequency produced by the model and inhibit the overexcitation of neurons. The kinetic process of Nav1.9 is similar to that of Nav1.8, and the changes in the activation curve and inactivation curve are consistent with the results of Nav1.8. These results suggest that the analgesic effect of N58A on trigeminal neuralgia is closely related to the regulation of Nav1.8 and Nav1.9 channels in TG neurons. In this experiment, morphine was selected as a positive drug, which had a very significant effect on analgesia, but there was no effect on Nav1.8 and Nav1.9 in the patch clamp test.

## 4. Conclusions

Scorpion analgesic peptide N58A has a significant analgesic effect on IoN-CCI model rats. Its analgesic effect may be achieved by reducing the phosphorylation level of MAPK pathway protein, reducing the expression of Nav1.8 and Nav1.9 protein in neuronal cell membrane, inhibiting Nav1.8 and Nav1.9 current, changing kinetic characteristics, and blocking the transmission of peripheral pain signals.

## 5. Materials and Methods

### 5.1. Animals and Treatments

Sprague-Dawley female rats were used in the experiment and weighed about 160–180 g. All experimental operations were performed in accordance with the regulations of the Animal Ethics Committee of Shenyang Pharmaceutical University, China (SCXK (Liao) 2015-0001). The animals were randomly divided into a control group, sham group, model group, positive drug (morphine, 4.0 mg/kg) group, and administration group (N58A), with 10 rats in each group. N58A was dissolved and diluted with normal saline to high (4.0 mg/kg), medium (2.0 mg/kg), and low (1.0 mg/kg) doses. On the 14th day after operation, the drug was injected into the tail vein, and different concentrations of N58A were given with a dose volume of 1.0 mL/100 g. The positive drug was the same as above. The sham group was given the same dose of normal saline.

### 5.2. IoN-CCI Model

This experiment used an improved trigeminal nerve IoN-CCI model [16]. Female SD rats were anesthetized by intraperitoneal injection of 3.5% chloral hydrate (1 mL/100 g). After anesthesia, between the beard pad and the eye of the female SD rat, starting from the end of the third row of the beard line near the eye, a 0.5 cm incision was made parallel to the orbit, and the muscle was bluntly separated to separate the infraorbital nerve branch. A 4.0 absorbable ligation thread was used to ligate the trigeminal ganglion. The degree of ligation was based on the slight narrowing of the nerves without affecting blood flow. After the operation, the incision was closed with sutures, and 80,000 U of penicillin injection was intraperitoneally injected to prevent postoperative infection in the rats.

### 5.3. Assessment of Pain Behavior after IoN-CCI Model

#### 5.3.1. Mechanical Allodynia

Mechanical allodynia threshold of the rat’s mouth and face: The rats were adaptively trained three days before the operation. After the operation, the von Frey electronic pain meter was used to detect the mechanical allodynia threshold of the right face of the rat. When the rat had any of the following behavioral changes, the intensity used was recorded, which was its mechanical pain threshold: (1) withdrawal response (that was, the head quickly retracted to dodge or blink) or (2) escape or offensive behavior (shown as fleeing, curling up, hiding the head, grabbing and biting irritants, etc.). The mechanical allodynia threshold recorded the day before the surgery was used as the baseline level for the test, which was then measured 3, 5, 7, 9, 11, 13, and 14 days after the surgery and 0.5, 2, and 4 h after administration.

#### 5.3.2. Thermal Withdrawal Latency

The rats were given adaptive training 3 days before operation, and the rats with calm response to training stimulation, and intact oral and facial hair and skin were selected for postoperative detection. Rats’ facial beard pads were exposed to high-intensity radiation beams, showing rapid head shaking or blinking, which can be thought of as an avoidance response. The time from the start of high-intensity radiation to the time when the rat demonstrates a dodge response is the incubation period of thermal stimulation response. In the experiment, a thermal pain meter with a thermal radiation beam temperature of 50 °C was used, and the light source was measured at 8 cm from the oral and facial test area.

#### 5.3.3. Spontaneous Asymmetrical Facial Grooming

The tested rats were placed alone in a transparent Plexiglass observation cage. After the rats adapted to the environment, the amount of spontaneous asymmetric self-grooming of the rat’s face within 5 min was observed. In a quiet environment, the experimenter gave the rat’s face uninterrupted stimulation. If the rat groomed and scratched the irritation site more than 3 times, it was considered that the rat had spontaneously and asymmetrically scratched its face. A scoring system was adopted for behavior evaluation; that is, each occurrence of spontaneous asymmetric scratching behavior was scored as 1 point. The score of autonomous behavior recorded the day before operation was used as the basic level of the test, and then detected at 14 days after operation and 0.5, 2, and 4 h after administration.

### 5.4. Gait Analysis

Gait analysis training was performed 7 days before operation, and the rats after gait analysis needed to adapt to the gait analyzer at a speed of 24 cm/s. Before the operation, the running of the rats on the gait analyzer at a speed of 24 cm/s was recorded and kept as a blank horizontal baseline. The animals were randomly divided into a sham group, sham administration group, and N58A group. N58A was diluted into a high dose (4.0 mg/kg) and low dose (2.0 mg/kg) with normal saline. According to the results of the behavioral experiments, when the pain reached the maximum effect on the 14th day of modeling, the drug was injected into the tail vein with a volume of 1.0 mL/100 g to give different concentrations of N58A, and the sham group was given the same dose of normal saline. Gait analysis was performed at 2 h and 24 h after administration.

### 5.5. Western Blot

After 2 h of administration, the experimental rats were killed, the total trigeminal nerve protein was extracted, and the histone concentration of each sample was determined by BCA method. The target protein was separated by SDS-PAGE electrophoresis, and the target protein and internal reference protein were transferred to PVDF membrane. After electroporation, the membrane was sealed in a vertical shaker with 5% skim milk powder-TBST sealing solution at room temperature for 1 h. After sealing, the membrane was taken and incubated in a configured primary antibody. After incubating in a vertical shaker at room temperature for 30 min, it was put into a refrigerator at 4 °C for the night. The next day, the membrane was removed and reheated for 30 min, then rinsed with TBST 3 times for 5 min each time, and then incubated with a second antibody at room temperature for 1 h. Finally, the developer was prepared and dripped evenly on the film, and after the full reaction, the gel imaging system was used to develop.

### 5.6. Acute Dissociation of TG Neurons

After complete anesthesia, the experimental SD rats were decapitated quickly, the skulls were cut, and a pair of trigeminal ganglia were quickly removed and put into the DMEM/F12 culture medium in an ice bath. The trigeminal ganglia were transferred to the super-clean platform, the fascia on the nerve surface was stripped with microscopic forceps, the tissue was cut into pieces as much as possible with ophthalmic scissors, and the tissue was transferred to the digestive juice and put into the culture box for full digestion. The trigeminal nerve tissue was transferred into a 5 mL glass centrifuge tube and gently blown 15 times with a glue dropper to disperse the tissue. It stood for 1 min, 0.3 mL of culture medium was added, it was gently blown upon for 3 min with a 200 μL liquid transfer gun, it rested for 1 min, and then the tissue suspension of TG neuron cells was sucked out after the tissue settled. The climbing plates were planted on 35 mm disposable Petri dishes coated with 0.01% polylysine, then 0.3 mL cell culture medium was added, and the above steps were repeated until the tissue disappeared. The Petri dishes were placed in a 37 °C incubator containing 5% CO_2_ and 95% O_2_ for cell attachment, and patch clamp experiments were carried out 2 h later.

### 5.7. Whole Cell Patch Clamp Experiment

Under the microscope, an electrode with a smooth tip and a resistance of 2 to 5 MΩ was selected. The “computer, digital-to-analog converter, amplifier, and Clampex 10.0 software” was opened in turn. A piece of climbing slice containing TG neurons was put into the bath and the cell surface was washed with extracellular sodium solution at 37 °C for 5 min to wash off the impurities on the cell surface. The round or oval neuronal cells with smooth surface, good stereoscopic sense, and strong refraction were moved to the center of the visual field. A directional thruster to slowly move the electrode tip close to the right side of the cell was used first, then a three-dimensional hydraulic micromanipulator was used to adjust the electrode tip to the top of the cell body, slowly pressing down the electrode and making it adhere to the cell until the current drops to 2/3 before the attachment. An appropriate amount of negative pressure was applied to seal the cell. When the sealing resistance was more than 1 GΩ, the sealing was considered to be intact. Then a certain negative pressure or a ZAP electric shock was applied to break the membrane, so that the intracellular fluid was connected with the electrode fluid to form a “whole-cell recording” mode, which was stable for 2–3 min. An appropriate protocol was selected for electrophysiological recording. The whole experiment was carried out at a room temperature of 25 °C.

### 5.8. Effects of N58A on Steady-State Activation and Inactivation Kinetics of Nav1.8 Currents

Nav1.8 currents were evoked to −50 mV over the course of 500 ms, followed by the application of voltage steps ranging from −50 to +50 mV in increments of 10 mV. The conductance G (V) was calculated according to G = I / (V − V_Na_), where V_Na_ is the reversal potential, V is the test pulse potential, and I is the current amplitude. Using G/G_max_ as ordinate and conditioned pulse voltage as Abscissa, the steady-state activation curves of Nav1.8 were fitted by the Boltzmann equation: G / G_max_= 1/{1 + exp[(V_1/2_− V)/k]}, where G_max_ is the maximum conductance, V_1/2_ is the membrane potential of half-maximal activation, and k is the slope factor.

In order to study the steady-state inactivation characteristics of Nav1.8 channels, the cells were first clamped at –100 mV, then the Nav1.8 inactivation current was recorded after a series of pre-pulses with a width of 500 ms, an increment of 10mV, followed by a test pulse of 0 mV, lasting 100 ms, and finally returning to –100 mV. Taking the ratio of the peak current to the maximum current measured under different conditioned stimulation pulses as the ordinate and the corresponding conditioned stimulus pulse voltage as the Abscissa, steady-state inactivation curves were determined with a Boltzmann fit of the data using *I / I_max_* = 1 / { 1 + exp[(*V − V_1/2_*) / *k* ] }, where *I* is the current amplitude, *I_max_* is the maximal current amplitude, *V* is the prepulse, *V*_1/2_ is the prepulse voltage at which the current amplitude is half maximum, and *k* is the slope factor.

### 5.9. Effects of N58A on Steady-State Activation and Inactivation Kinetics of Nav1.9 Currents

The Nav1.9 currents were evoked by a single voltage step at –35 mV from a holding potential of –100 mV. The wave width was 700 ms, and the increment was 5 mV, and finally returned to the clamp voltage of –100 mV. According to the protocol, a series of Nav1.9 currents were recorded. The steady-state activation curve of Nav1.9 was drawn by fitting the Boltzmann equation.

In order to study the steady-state inactivation characteristics of the Nav1.9 channel, a series of pre-pulses with a width of 700 ms and an increment of 5 mV, from –100 mV to –35 mV, were given, followed by a test pulse of –35 mV, lasting 100 ms, and finally returning to a clamp voltage of –100 mV, from which the inactivation current of Nav1.9 could be recorded.

### 5.10. Statistical Analysis

The experimental data were analyzed and plotted by SPSS 22.0 and Graphpad Prism 7.0 software. The experimental data are expressed as mean ± SEM error. One-way ANOVA was used for the comparison between groups. The LSD test was used when the variance was homogeneous, and Dunnett’s T3 test was used when the variance was uneven. When *p* < 0.05, it was considered that there was a statistically significant difference.

## Figures and Tables

**Figure 1 toxins-13-00357-f001:**
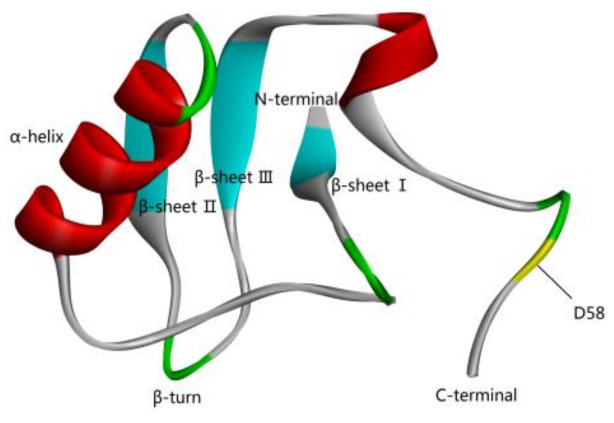
Tertiary structure of scorpion analgesic peptide N58A.

**Figure 2 toxins-13-00357-f002:**
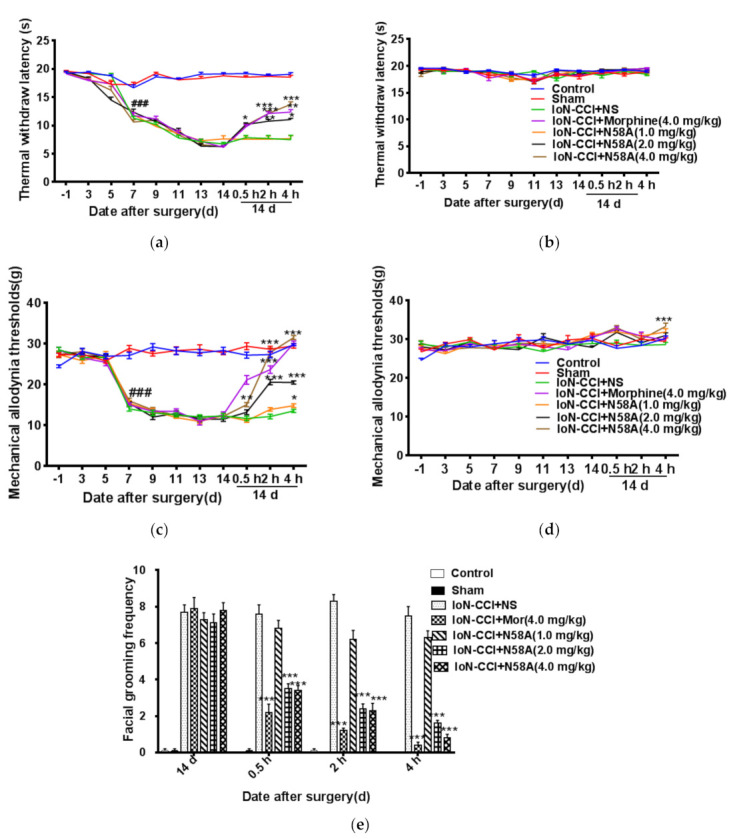
Behavioral changes after the IoN-CCI model in rats. (**a**) Changes in thermal hyperalgesia latencies in groups on the ipsilateral side of the rat. The middle-dose and high-dose groups showed significant analgesic effect 0.5 h after administration, and the analgesic effect was similar to that of morphine. (**b**) Changes in thermal hyperalgesia latencies in groups on the contralateral side of the rat. (**c**) Changes in mechanical allodynia thresholds in groups on the ipsilateral side of the rat. The middle-dose and high-dose groups showed significant analgesic effect 0.5 h after administration, and the analgesic effect was similar to that of morphine in a dose-dependent manner. (**d**) Changes in mechanical allodynia thresholds in the contralateral side of rats. (**e**) Changes in scores of spontaneous asymmetrical facial grooming on the rats’ right whisker pads caused by N58A and morphine infusion. It is obvious that N58A could inhibit asymmetrical facial grooming significantly in dose and time. All data are presented as mean ± SEM. *n* = 10. ### *p* < 0.001 compared with sham group. * *p* < 0.05, ** *p* < 0.01, *** *p* < 0.001 compared with IoN-CCI + NS group.

**Figure 3 toxins-13-00357-f003:**
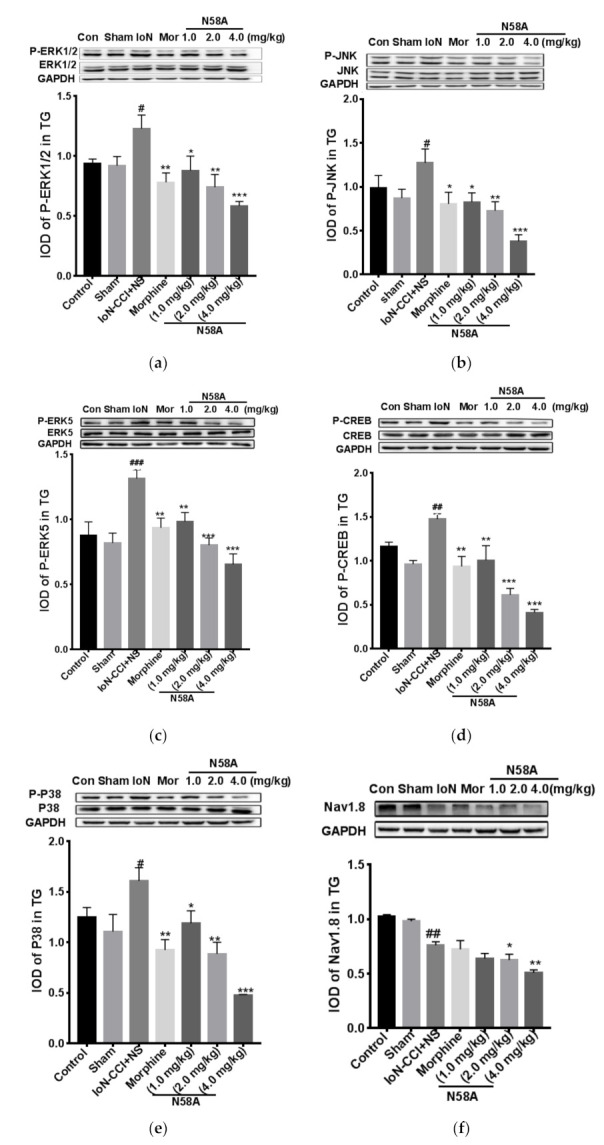
The effects of N58A on protein phosphorylation of MAPK and ERK5/CREB pathways and expression of Nav1.8 and Nav1.9 proteins. (**a**) There are significant decreases in the phosphorylation levels of ERK1/2 after treating with N58A. (**b**) There are significant decreases in the phosphorylation levels of JNK after treating with N58A. (**c**) There are significant decreases in the phosphorylation levels of ERK5 after treating with N58A. (**d**) There are significant decreases in the phosphorylation levels of CREB after treating with N58A. (**e**) There are significant decreases in the phosphorylation levels of P38 after treating with N58A. (**f**) There are significant decreases in the expression level of Nav1.8 after treating with 2.0 mg/kg and 4.0 mg/kg N58A. (**g**) There is a significant decrease in the expression level of Nav1.9 after treating with 4.0 mg/kg N58A. All data are presented as mean ± SEM. *n* = 3. # *p* < 0.05 compared with sham group. * *p* < 0.05, ** *p* < 0.01, *** *p* < 0.001 compared with IoN-CCI+NS group. # *p* < 0.05, ##*p* < 0.01, ### *p* < 0.001 compared with sham group.

**Figure 4 toxins-13-00357-f004:**
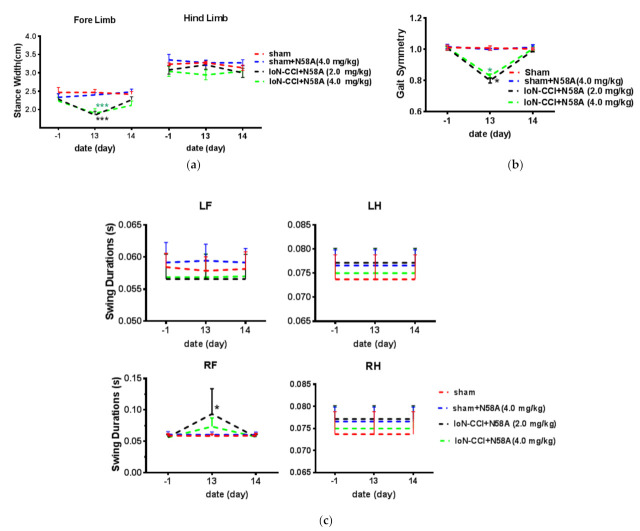
The effect of N58A on motor function of IoN-CCI rats. (**a**) Effects of N58A on stance width following the IoN-CCI model established in rats. On the 13th day after the establishment of the model, the width of the forefoot touchdown support in the model group was significantly lower than that in the sham group. (**b**) Effects of N58A on gait symmetry following the IoN-CCI model established in rats. (**c**) Effects of N58A on swing time after the IoN-CCI established in rats. All data are presented as mean ± SEM, *n* = 6. ** p* < 0.05, **** p* < 0.005 compared with sham group on day 13.

**Figure 5 toxins-13-00357-f005:**
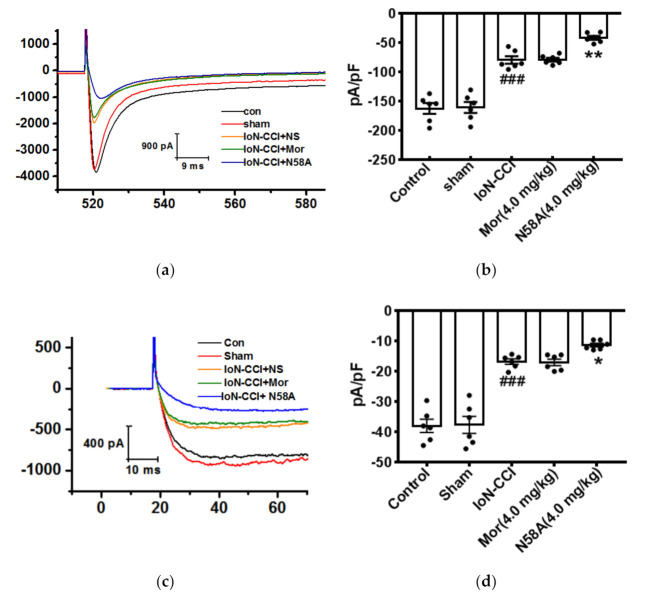
Changes in Nav1.8 and Nav1.9 peak current densities in TG neurons after IoN-CCI treatment and N58A injection. (**a**) The Nav1.8 currents were evoked by a depolarization voltage step from –50 mV to +50 mV in 10 mV increments from a holding potential of –110 mV. After IoN-CCI surgery, the peak sodium current density mediated by Nav1.8 was significantly decreased by 51.1%; 4.0 mg/kg N58A inhibited Nav1.8 current by 48.0%. (**b**) Effect of N58A on Nav1.8 current density in IoN-CCI rat TG neurons. (**c**) The Nav1.9 currents were evoked by a single voltage step at –35 mV from a holding potential of –100 mV. (**d**) Effect of N58A on Nav1.9 current density in IoN-CCI rat TG neurons. After IoN-CCI surgery, the sodium current density mediated by Nav1.9 was decreased by 56.0%; 4.0 mg/kg N58A inhibited Nav1.9 current by 30.0%. All data are shown as mean ± SEM. *n* = 6. *### p* < 0.001 compared with sham group. *** p* < 0.01 compared with IoN-CCI + NS group. * *p* < 0.05.

**Figure 6 toxins-13-00357-f006:**
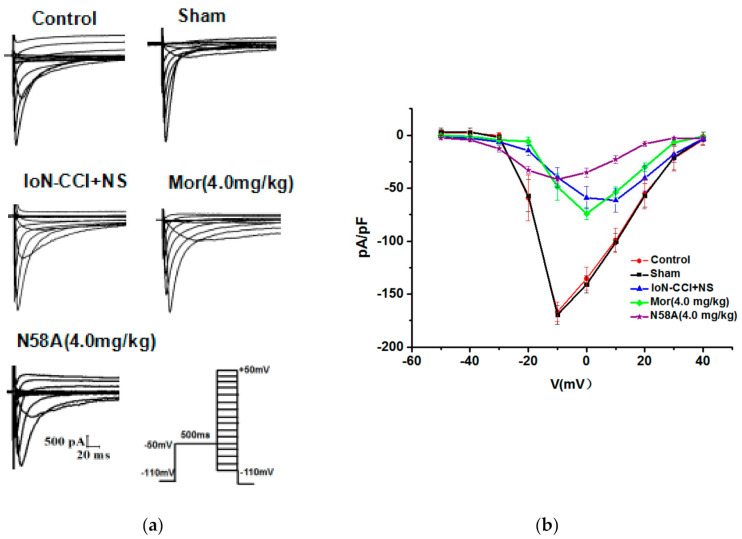
Changes in Nav1.8 and Nav1.9 currents I-V and steady-state activation curves in TG neurons after IoN-CCI treatment and N58A injection. (**a**) Representative I-V curve family of currents recorded in the presence of 1000 nM TTX. Cells were depolarized to a variety of potentials (−50 mV to +50 mV) from a holding potential of –110 mV, to elicit Nav1.8 currents (*n* = 6). (**b**) I-V curves of Nav1.8 currents. (**c**) Representative I-V curve family of currents recorded in the presence of 1000 nM TTX. Cells were depolarized to a variety of potentials (−80 mV to –35 mV) from a holding potential of –100 mV, to elicit Nav1.9 currents (*n* = 6). (**d**) I-V curves of Nav1.9 currents. (**e**) Changes in Nav1.8 steady-state activation curves in TG neurons after IoN-CCI treatment and N58A injection. IoN-CCI surgery shifted the steady-state activation curve in a depolarization direction; 4.0 mg/kg N58A shifted the steady-state activation curve in a hyperpolarizing direction (*n* = 6). (**f**) Changes in Nav1.9 steady-state activation curves in TG neurons after IoN-CCI treatment and N58A injection. IoN-CCI surgery shifted the steady-state activation curve in a depolarization direction; 4.0 mg/kg N58A shifted the steady-state activation curve in a hyperpolarizing direction (*n* = 6). All data are presented as mean ± SEM. # *p* < 0.05 compared with sham group. *** *p* < 0.005 compared with IoN-CCI+NS group.

**Figure 7 toxins-13-00357-f007:**
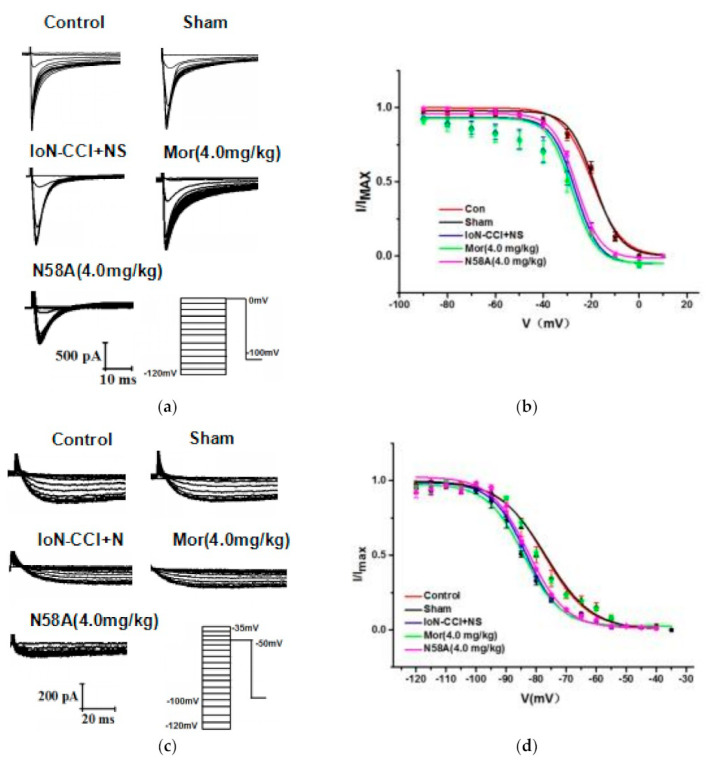
Changes in Nav1.8 and Nav1.9 steady-state inactivation curves after IoN-CCI treatment and N58A injection. (**a**) Nav1.8 inactivation currents were generated by 10 mV progressive steps between –120 and 0 mV from a holding potential of –100 mV. (**b**) Nav1.8 steady-state inactivation curves. IoN-CCI surgery shifted the steady-state inactivation curve in a hyperpolarizing direction; 4.0 mg/kg N58A shifted the steady-state inactivation curve in a depolarization direction (*n* = 6). (**c**) The protocol to elicit Nav1.9 inactivation currents starting from a holding potential of −100 mV, applying conditioning pulses ranging from −110 to –35 mV in increments of 5 mV, and applying a test pulse at −50 mV. (**d**) Nav1.9 steady-state inactivation curves. IoN-CCI surgery shifted the steady-state inactivation curve in a hyperpolarizing direction; 4.0 mg/kg N58A shifted the steady-state inactivation curve in a depolarization direction (*n* = 6).

## Data Availability

Data is contained within the article or Appendix A.

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
