# Peer review of "N58A Exerts Analgesic Effect on Trigeminal Neuralgia by Regulating the MAPK Pathway and Tetrodotoxin-Resistant Sodium Channel"

_toxins, 2021, doi:10.3390/toxins13050357_

Round 1

Reviewer 1 Report

The authors describe the analgesic effects of a scorpion peptide, that targets voltage-gated sodium channels, on trigeminal neuralgia model. The data are quite interesting, however I found that the paper needs to be properly rewritten . The description of voltage-clamp stimulation protocols are not well described in the material and method section. The figures are all of poor quality (check the size of characters). Morever, the figures should be improved (choice of colors, legends...). The sequence of the peptide should be deposited in databank.

Please check English.

Altogether, the data are quite good, but the article is not well written. It would take me a long time to describe all points to reconsider for a suitable manuscript for the journal Toxins. 

In all figures, the legends are written with a too small size. They are very difficult to read. The grey-scale colors are not really helpful.

Figures 2, 3 and 5: Is it possible to show the data as scatter plots with bar? It would be better for illustrating the data and evaluate their relevance.

To record correctly Nav1.9 channels, it would be better to hold the potential at -120 mV. However, to record Nav1.8, the best is to hold at -70 mV. At this potential, Nav1.9 is totally inactivated. Check the litterature (ex : Wu et al. 2012. J Clin Invest. 122(4):1306–1315).

Figure 5 : the statistical analysis are not enough described.

The voltage-dependancy of activation is altered by N58A peptide as shown in figure 6. However, the statistical analysis was not shown. 

Figure 7 : it is very difficult to see the line of the inactivation curves in all conditions. The 2-pulse stimulation protocol should be described (pulse duration) and please show the currents seen with the 2nd pulse.

Reviewer 2 Report

This manuscript describes the experimental results of applying the scorpion toxin N58A to rat neuronal models and patch clamp assays on NaV channels, in particular 1.8 and 1.9. Sodium channels are involved in pain reactions and so finding novel analgesics is a key research area for pain control.

The major flaw with this manuscript is it does not cite the original discovery of the peptide. I cannot locate the peptide in Uniprot, find a structure associated in the PDB (which has been reproduced in the manuscript), nor locate any previous studies characterising N58A. There is no mention of what scorpion it came from nor how the venom and peptide were isolated. The missing referencing for all of the previous work on “N58A” means I cannot verify any of this work.

The structure does not exist in the Protein Database, the inference in L 80 is that the structure has been clearly described but absolutely no reference to how or by whom. The sequence itself described in the manuscript matches a scorpion knottin family which includes defensins, Alpha and Beta scorpion toxins in the PFAM database, but again the sequence has not been deposited in any databases that I can find.

If the peptide is newly discovered by the authors then extensive revision of the manuscript is required to include all the methods applicable to the collection of the animals, isolation of the peptide from the venom. Until these issues have been rectified I am unable to recommend the paper for publication nor can I review the manuscript correctly until these issues have been addressed.

Round 2

Reviewer 1 Report

The authors have followed my recommandation and the manuscript has been improved. I think it will provide an interesting and convincing work on N58, a peptidyl scorpion toxin that exhibits analgesic effects. In my opinion, this article will be useful for the readership of "Toxins".

Author Response

Thank you very much for your comments and suggestions. We have further modified the manuscript, and the language in the manuscript has also been checked and corrected.

Reviewer 2 Report

I recommend for publication however I still feel the scorpion that the peptide was derived from needs to be stated in the introduction.

In addition as it is a new peptide then this needs to be lodged and accession number included in the manuscript prior to publication. The peptide sequence does not need to be released for public prior to publication. 
